# Structural insights into the nucleic acid remodeling mechanisms of the yeast THO-Sub2 complex

**Sandra K Schuller†‡, Jan M Schuller†§, J Rajan Prabu, Marc Baumgärtner, Fabien Bonneau, Jérôme Basquin, Elena Conti\***

Department of Structural Cell Biology, Max Planck Institute of Biochemistry, Munich, Germany

**\*For correspondence:**
conti@biochem.mpg.de

†These authors contributed equally to this work

**Present address:** ‡Center for Synthetic Microbiology (SYNMIKRO) and Max Planck Institute for Terrestrial Microbiology, Marburg, Germany; §Center for Synthetic Microbiology (SYNMIKRO) and Department of Chemistry, Philipps University Marburg, Marburg, Germany

**Competing interests:** The authors declare that no competing interests exist.

**Abstract** The yeast THO complex is recruited to active genes and interacts with the RNA-dependent ATPase Sub2 to facilitate the formation of mature export-competent messenger ribonucleoprotein particles and to prevent the co-transcriptional formation of RNA:DNA-hybrid-containing structures. How THO-containing complexes function at the mechanistic level is unclear. Here, we elucidated a 3.4 Å resolution structure of *Saccharomyces cerevisiae* THO-Sub2 by cryo-electron microscopy. THO subunits Tho2 and Hpr1 intertwine to form a platform that is bound by Mft1, Thp2, and Tex1. The resulting complex homodimerizes in an asymmetric fashion, with a Sub2 molecule attached to each protomer. The homodimerization interfaces serve as a fulcrum for a seesaw-like movement concomitant with conformational changes of the Sub2 ATPase. The overall structural architecture and topology suggest the molecular mechanisms of nucleic acid remodeling during mRNA biogenesis.

## Introduction

The biogenesis of eukaryotic mRNAs in the cell nucleus is an elaborate, multi-step process. As the nascent transcript is synthesized by RNA polymerase II from the corresponding DNA template, it undergoes several chemical modifications, including 5' capping, splicing and 3'-end polyadenylation. Simultaneously, the nascent transcript associates with a cohort of complementary proteins to form a mature messenger ribonucleoprotein particle (mRNP) (*Wende et al., 2019*; *Xie and Ren, 2019*; *Müller-McNicoll and Neugebauer, 2013*; *Meinel and Sträßer, 2015*; *Singh et al., 2015*; *Wegener and Müller-McNicoll, 2018*). The mature mRNP is then shuttled through the nuclear pore to the cytoplasm via specific export factors (*Moore and Proudfoot, 2009*; *Müller-McNicoll and Neugebauer, 2013*; *Rondón et al., 2010*; *Wende et al., 2019*; *Xie and Ren, 2019*). The individual steps in this process are orchestrated by multi-protein complexes. Among them is the transcription-export (TREX) complex, an evolutionary conserved complex containing a multi-protein assembly (transcription-dependent hyperrecombination complex THO), an RNA helicase (Sub2 in yeast/UAP56 in human), and an hnRNP-like protein (Yra1 in yeast/Ref/Aly in human) (*Heath et al., 2016*; *Meinel and Sträßer, 2015*; *Strässer and Hurt, 2001*; *Strässer et al., 2002*; *Xie and Ren, 2019*). Sub2 and Yra1 interplay with the mRNA-export factor Mex67-Mtr2 (TAP-p15 in human) (*Heath et al., 2016*; *Strässer and Hurt, 2001*; *Xie and Ren, 2019*). As the name implies, the TREX complex is believed to play pivotal roles in linking the transcription and nuclear export steps in the mRNP biogenesis pathway, but the molecular mechanisms remain unclear.

The THO complex was first identified in *Saccharomyces cerevisiae* and named after the phenotypes observed upon mutations of its constituent subunits, most notably an increase in recombination events connected to the accumulation of R loops during transcription (*Aguilera and Klein, 1988*; *Chávez et al., 2000*; *Chávez et al., 2001*; *Heath et al., 2016*; *Huertas and Aguilera, 2003*;

*Piruat and Aguilera, 1998*). R-loops can form behind the elongating RNA polymerase II when the nascent transcript erroneously reanneals with the DNA template to create a three-stranded nucleic acid structure consisting of an RNA:DNA hybrid and an unpaired single DNA strand. R loops are particularly prominent in discrete areas of the genome and can lead to recombination events and, eventually, genomic instability (*Santos-Pereira and Aguilera, 2015*; *Skourti-Stathaki et al., 2014*). Among the subunits of the yeast THO complex, the evolutionary conserved Tho2 and Hpr1 have the most drastic impact on mutation-induced phenotypes epitomized by a DNA recombination rate of about 3000 times that found in wild-type cells (*García-Rubio et al., 2008*; *Chávez et al., 2000*). The mechanisms with which Tho2, Hpr1, and the other yeast THO complex core subunits (Mft1, Thp1, and Tex1) protect against R loop formation are thought to involve the unwinding activity of the Sub2 helicase (*Luna et al., 2019*). Indeed, THO enhances the ATPase properties of Sub2 (*Ren et al., 2017*), but there is currently no high-resolution structure of this complex to allow a mechanistic understanding (*Ren et al., 2017*).

Yeast Sub2 is a conserved RNA-dependent ATPase of the DEAD-box family, a group of enzymes known to destabilize and unwind RNA duplexes (*Linder and Jankowsky, 2011*; *Xie and Ren, 2019*). Biochemical studies have shown that Sub2 interacts tightly with THO both in vivo and in vitro (*Jimeno et al., 2002*; *Luna et al., 2019*; *Peña et al., 2012*; *Ren et al., 2017*; *Strässer et al., 2002*). Furthermore, THO mutant alleles are synthetic lethal with a temperature-sensitive mutant of *sub2* causing a fast onset of nuclear poly(A) RNA accumulation, a hallmark of defective mRNA export (*Jimeno et al., 2002*; *Strässer et al., 2002*; *Wegener and Müller-McNicoll, 2018*). The mRNA export functions of Sub2 are mediated by its interacting partner Yra1, a non-shuttling hnRNP-like protein (*Heath et al., 2016*; *Strässer and Hurt, 2001*; *Stutz et al., 2000*; *Xie and Ren, 2019*). The propensity of Yra1 to bind genomic areas that are prone to R-loop formation (*García-Rubio et al., 2008*) is underscored by its ability to bind both RNA and DNA (*Abruzzi et al., 2004*). Additionally, Yra1 harbors potent RNA:RNA annealing activity (*Portman et al., 1997*). Data from both yeast and human studies have converged on the notion that THO binds Sub2, which in turn recruits Yra1 to form the TREX complex (*Heath et al., 2016*; *Strässer et al., 2002*; *Wende et al., 2019*; *Zenklusen et al., 2002*). This series of events is remarkably conserved in the human orthologues, the DNA–RNA helicase UAP56 (*DDX39B*) and the RNA-binding factor Aly/Ref (*ALYREF*) (*Heath et al., 2016*; *Luo et al., 2001*; *Strässer et al., 2002*; *Wende et al., 2019*). In this study, we identify the molecular mechanisms utilized by yeast THO to activate the unwinding properties of the Sub2 ATPase, thereby intimating how THO-containing complexes can fulfil their functions in R loop prevention and mRNP formation.

## Results and discussion

### The *S. cerevisiae* THO complex is a dimer upon in vitro reconstitution and upon endogenous purification

*S. cerevisiae* Tho2 (1597 aa, 184 kDa), Hpr1 (752 aa, 88 kDa), Mft1 (392 aa, 45 kDa), and Thp2 (261 aa, 30 kDa) are all expected to contain folded N-terminal regions that encompass more than two-thirds of their polypeptide chain and to be primarily α-helical in secondary structure as predicted by the program Phyre2 (*Kelley et al., 2015*; *Figure 1A*). Tex1 (422 aa, 47 kDa) is the only non-helical subunit of the THO complex and contains a β-propeller domain (*Kelley et al., 2015*; *Figure 1A*). Since, with the exception of Tex1, it was not possible to accurately predict the domain boundaries between the structured and unstructured regions of THO complex subunits, we co-expressed them as full-length proteins in insect cells. The recombinant THO complex was purified to homogeneity using affinity and size-exclusion chromatography (*Figure 1B*). Full-length Sub2 (446 aa, 50 kDa) was individually expressed, purified, and added to the THO complex. The THO-Sub2 assembly eluted as a single peak by size-exclusion chromatography, albeit at an earlier elution volume than expected for a globular mass of ~450 kDa and rather consistent with the presence of a dimer (*Figure 1B*).

To assess the presence of THO dimerization in a cellular environment, we integrated a C-terminal tandem-affinity purification tag (Twin-strep-3C-Protein-A) at the endogenous HPR1 locus in BY4741 (haploid) and BY4743 (diploid) *S. cerevisiae* strains. The haploid strain produces only a tagged version of Hpr1, whereas the diploid strain retains both tagged and untagged versions. The native complex was isolated from both yeast strain extracts by tandem affinity purification (*Figure 1C*). An

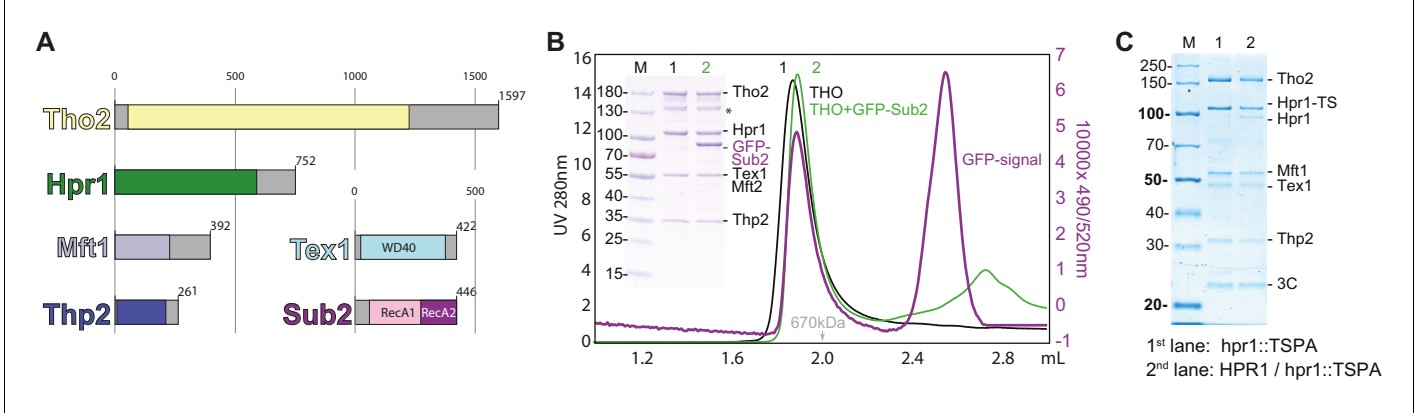

**Figure 1.** Biochemical reconstitution and native isolation of *Saccharomyces cerevisiae* THO-Sub2. (**A**) Domain organization of Tho2 (yellow), Hpr1 (green), Mft1 (light blue), Thp2 (dark blue), Tex1 (cyan), and Sub2 (with the RecA1 domain in pink and the RecA2 domain in purple). Gray parts are not resolved in the structural analysis described in the paper and correspond to regions predicted to be mainly unstructured. (**B**) Analytical size-exclusion chromatography of THO with/without GFP-Sub2. The co-elution of GFP-Sub2 was monitored by fluorescence at excitation 490 and emission 520 nm. The green line is the 280 nm absorbance signal of THO with GFP-Sub2; the purple line is the fluorescence signal of THO with GFP-Sub2. The asterisk indicates a Tho2 degradation product. (**C**) Tandem affinity isolation of native THO-containing complexes from haploid and diploid yeast. A single allele of HPR1 was tagged C-terminally with a Twin-Strep-3C-Protein-A tag (TSPA) in BY4741 (haploid) or BY4743 (diploid). Eluates resulting from IgG-affinity followed by Strep-tactin (IBA) affinity purification were analyzed on 12% SDS-PAGE stained with Instant Blue (Expedeon). M, molecular weight marker; Hpr1-TS, Hpr1 twin strep.

additional band corresponding to untagged Hpr1 was also observed in the complex originating from the diploid strain, consistent with the presence of a dimer in a physiological context.

## Cryo-electron microscopy (cryo-EM) analysis of a yeast THO complex bound to Sub2

For cryo-EM single-particle analysis, we stabilized the THO-Sub2 assembly with the mild crosslinker BS3 to obtain a homogeneous particle distribution suitable for high-resolution reconstructions. After iterative-rounds of particle sorting and refinements, the reconstruction revealed the presence of an asymmetric dimer, with a well-ordered protomer resolved to a resolution of 3.7 Å and a more flexible protomer with a resolution spread of 3.8–7 Å, referred to as rigid and flexible protomers, respectively (*Figure 2A* and *Figure 2—figure supplement 1*; *Supplementary file 1*). Focusing on the classification of the entire dataset on the rigid protomer, we improved the density map to 3.4 Å resolution (*Figure 2—figure supplement 2*), allowing us to build the atomic model of the pentameric THO protomer de novo (*Figure 2B* and *Figure 2—figure supplement 3*; *Supplementary file 1*). The final model includes the large N-terminal structured regions of the five THO-complex subunits as well as a prominent low-complexity sequence of Hpr1 (residue 501–675) (*Figure 1A*). The densities for Sub2 were modeled using the RecA domains from the reported high-resolution crystal structure (*Ren et al., 2017*). The density of the flexible protomer was interpreted using the structure of the rigid protomer (*Figure 2B*). The THO-Sub2 homodimer has a complex intertwined architecture (*Figure 2B and C*). Maximum-likelihood variance analysis of the cryo-EM data revealed that the significant nonuniform distribution of resolution resulted from the remarkable dynamic character of this assembly: the two protomers swivel with respect to each other, alternating the opening and closing of each side of the homodimer (referred to as proximal and distal sides; *Figure 2D*; *Video 1*). Since the sample we characterized by cryo-EM did not contain an energy source, these fluctuations likely reflect thermal motions. In support of our cryo-EM reconstruction (*Figure 2*), our findings are consistent with a previous negative-stain analysis of a native THO complex purified from yeast (*Peña et al., 2012*; *Figure 2—figure supplement 4*). Although a previous THO model based on 6 Å resolution crystallography data (*Ren et al., 2017*) appears different from our structure at first glance, the differences can be reconciled by reinterpreting the crystal lattice (*Figure 2—figure supplement 4*).

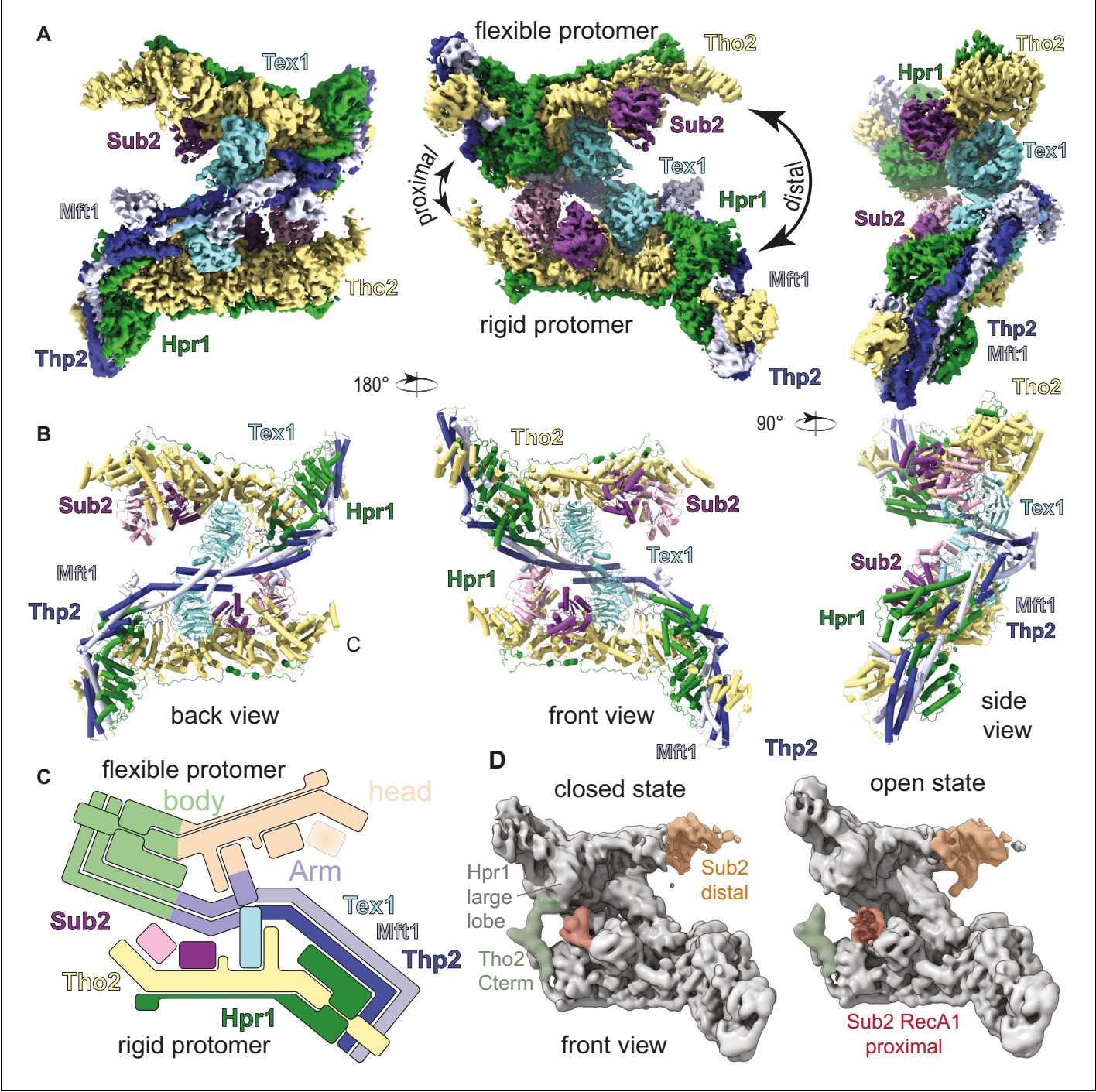

**Figure 2.** Cryo-electron microscopy (cryo-EM) reconstruction of *Saccharomyces cerevisiae* THO-Sub2 homodimer. (**A**) Segmented cryo-EM reconstruction of the THO-Sub2 dimer. Three different views are shown; proteins and domains are colored as in *Figure 1A*. Features discussed in the text are indicated, including the proximal and distal sides of the asymmetric homodimer, the rigid and flexible protomers, as well as the 'head and 'body' of each protomer. (**B**) Cartoon representation of the structure, shown in the same orientations and colors. Helices are rendered as solid cylinders. (**C**) Schematic representation of the THO-Sub2 complex architecture based on the cryo-EM structure. (**D**) Two frames of the raw cryo-EM data outputs from the variance analysis shown in *Video 1*. The Tho2 C-termini of the two protomers are shown in orange and green. Different conformations are adopted as the dimer switches the proximal and distal sides.

The online version of this article includes the following figure supplement(s) for figure 2:

**Figure supplement 1.** Cryo-electron microscopy analysis.

**Figure supplement 2.** High-resolution THO-Sub2 density.

*Figure 2 continued on next page*

*Figure 2 continued*

**Figure supplement 3.** Model quality.
**Figure supplement 4.** Comparison of our cryo-electron microscopy structure with previous structural studies.

## Overall structure of yeast THO-Sub2 protomer

In each THO protomer, the proteins Tho2 and Hpr1 together create the platform of the complex (*Figure 3A*). The platform is about 200 Å in length, with an oval 'head' and with an extended 'body' formed almost exclusively by alpha-helices. The 'head' is formed by helical repeats of Hpr1 and a loosely structured region of Tho2. Conversely, the 'body' is formed by an extended array of helical repeats of Tho2 (mostly bi-helical HEAT repeats [*Andrade et al., 2001*]) and a largely unstructured segment of Hpr1. The two smallest THO proteins, Mft1 and Thp2, interact to form the 'arm' of the complex, a striking 200 Å long coiled-coil structure that is oriented diagonally with respect to the Tho2-Hpr1 platform. The Tho2-Hpr1 platform can arbitrarily be dissected into five contiguous modules (module-1 to -5) (*Figure 3B*).

## The 'head' of the Tho2-Hpr1 platform binds Mft1-Thp2

The 'head' of each protomer comprises the first two Tho2-Hpr1 modules and forms the binding platform for Mft1-Thp2 (*Figure 3A*). Module-1 consists of the N-terminal portions of Tho2-Hpr1 and recognizes the N-terminal portion of the Mft1-Thp2 coiled coil (*Figure 3C* and *Figure 3—figure supplement 1*). Here, a four-helix bundle, consisting of a pair of helices from each Mft1 and Thp2, is sandwiched between a small HEAT-repeat lobe of Hpr1 and a composite helical fold formed by intercalating helices of Tho2 and Mft1 (*Figure 3A*).

Module-1 connects to module-2 via an extended segment of Tho2 and a large network of conserved interactions (*Figure 3—figure supplements 1–3*). Module-2 contains a large helical lobe of Hpr1 packed against the beginning of the helical array of the 'body' (the Tho2 HEAT1-2 bihelical repeats) (*Figure 3A*). Module-2 recognizes the middle portion of Mft1-Thp2, a two-helix coiled coil that is clamped within the two halves of the Hpr1 large helical lobe (*Figure 3D* and *Figure 3—figure supplement 3*). Although Mft1 and Thp2 share little conservation with their metazoan orthologues, the surfaces to which they bind on Tho2-Hpr1 contain evolutionary conserved residues (*Figure 3—figure supplements 1–3*), suggesting that the two predicted orthologues in the metazoan complex (THOC5 and THOC7) may share similar architectural features. In module-2, the Hpr1 large lobe and Mft1 also feature well-structured loops that latch on to module-3 (*Figure 3B and D*).

## The 'body' of the Tho2-Hpr1 platform binds Tex1 and Sub2

The 'body' of the complex contains the other three modules organized around the helical repeat array of Tho2 and the extended region of Hpr1 encompassing residues 501–605 (*Figure 3A*). Module-3 is a MIF4G-like fold with five Tho2 helical repeats (HEAT 3–7) lined underneath by Hpr1 residues 501–522 (*Figure 3E* and *Figure 3—figure supplement 2*). This module is characterized by two prominent insertions. A Tho2 β-hairpin insertion (between HEAT 5 and 6) forms a 50 Å protrusion that extends longitudinally on the concave side of the domain (*Figure 3E*). This insertion is buttressed along its length by Mft1 and Hpr1 loops from module-2 and reaches with its tip the C-terminal coiled-coil portion of Mft1-Thp2 (*Figure 3B*). Module-3 provides the Tex1-binding platform: the curved surface of the Tex1 β-propeller (blades 4 and 5) binds the concave side of the MIF4G-like fold (at Tho2 HEAT 6 and 7) with conserved interactions (*Figure 3—figure supplement 1F*). In addition,

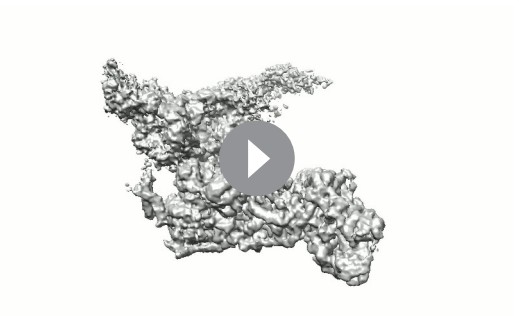

**Video 1.** Dynamic character of the complex extracted from cryo-electron microscopy data. Variance analysis of the THO-Sub2 complex structure, showing the swiveling motion of the two protomers with respect to each other.
https://elifesciences.org/articles/61467#video1

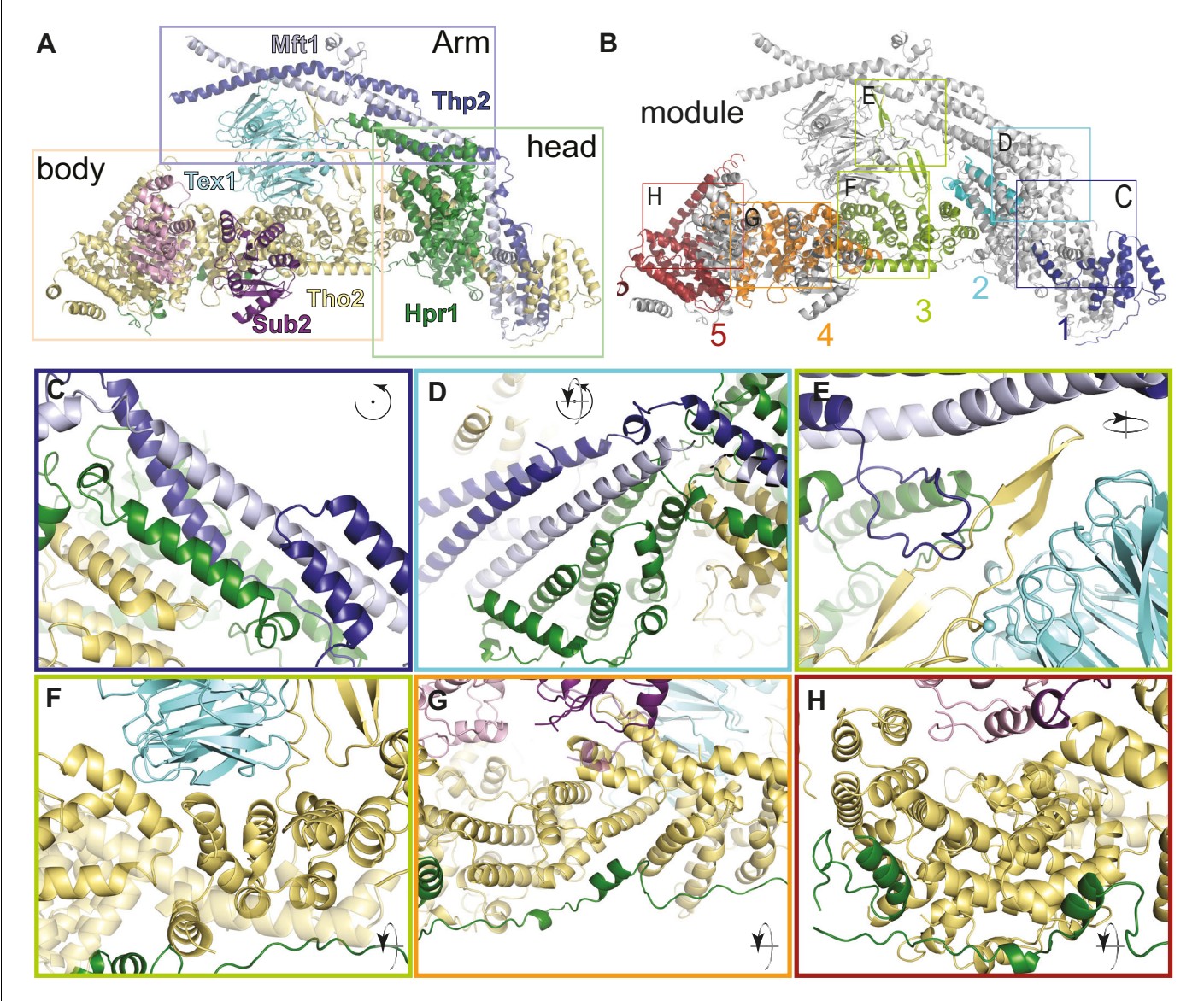

**Figure 3.** THO complex is built from intertwined conserved interactions. (A) Front view of the THO-Sub2 rigid protomer shown as a cartoon backbone representation. (B) Same view as A with the five modules of the Tho2-Hpr1 platform in different colors. The rectangles highlight the position of the zoom-ins shown in panels C–H. (C–H) Zoom-in views showing the intermolecular interactions between different subunits of a THO-Sub2 protomer as discussed in the text. The cartoon representations show the molecule either in the same view as panel A or after the indicated rotation. Interactions are shown between: (C) Tho2-Hpr1 module-1 and the N-terminal portion of the Mft1-Thp2 coiled-coil; (D) module-2 and central portion of the Mft1-Thp2 coiled-coil; (E) module-3 interactions: Tho2 β-hairpin, bottom surface of Tex1, and loop from the C-terminal Mft1-Thp2 coiled-coil region; (F) Tho2-Hpr1 module-3 and curved surface of Tex1 β-propeller; (G) Tho2-Hpr1 module-4 and Sub2 RecA2 domain; (H) Tho2-Hpr1 module-5 and Sub2 RecA1 domain. The online version of this article includes the following figure supplement(s) for figure 3:

**Figure supplement 1.** Structural features of the THO complex.
**Figure supplement 2.** Tho2 structure-based sequence alignment.
**Figure supplement 3.** Hpr1 structure-based sequence alignment.
**Figure supplement 4.** Tex1 structure-based sequence alignment.
**Figure supplement 5.** Sub2 structure-based sequence alignment.

the bottom surface of the Tex1 β-propeller interacts with the Tho2 β-hairpin (*Figure 3E and F* and *Figure 3—figure supplements 1*, *2,* and *4*). The other prominent insertion in module-3 is a long intra-repeat segment (at Tho2 HEAT 3) that packs along the edge of the MIF4G-like repeats and continues with a loop (referred to as the 'handle') latching on to module-4 (*Figure 3B, F, and G*).

The Sub2-binding platform in the 'body' of the complex is formed by module-4 and module-5 (*Figure 3A*). Module-4 comprises a MIF4G-like fold of Tho2 flanked by tri-helical motifs at both ends (referred to as ARM 8 and ARM 12 and 13) and lined on the convex side by conserved interactions with the Hpr1 extended region spanning residues 523–545 (*Figure 3G* and *Figure 3—figure supplements 1–3*). The RecA2 domain of Sub2 interacts with the concave side of this MIF4G-like fold (at Tho2 ARM 8 and HEAT 9) and contacts the 'handle' from module-3 (*Figure 3G* and *Figure 3—figure supplements 1* and *5*). Module-4 also contains a Tho2 helical insertion (between HEAT 11 and ARM 12) that latches on to module-5. In module-5, Tho2 has a V-shaped fold characterized by a larger side buttressed underneath by Hpr1 residues 546–605 and a smaller side also containing the helical insertion from module-4 (*Figure 3B and H*). In the rigid protomer, the RecA1 domain of Sub2 binds the concave surface of module-5 (Tho2 HEAT 16) and the helical insertion from module-4 (*Figure 3H*). In the flexible protomer, the RecA1 domain does not have ordered density. Another difference between the two protomers is at the ends of the Mft1-Thp2 pairs, which change course roughly at the point where the Tho2 β-hairpin contacts the coiled coil (*Figure 4—figure supplement 1*). Both differences relate to the homodimerization features of the complex, described below.

## THO homodimerization is mediated by Thp2-Mft1 and Tex1

In the THO homodimer, the two Tho2-Hpr1 platforms are arranged in an antiparallel fashion, i.e. the 'head' of one protomer faces the 'body' of the other (*Figure 2*). The coiled-coil structures of the two Mft1-Thp2 'arms' extend from opposite ends toward each other forming a chevron-like structure, with the two Tex1 protomers positioned below. The major homodimerization interface in the THO complex is mediated by Mft1-Thp2. The tips of the C-terminal coiled-coil portions of the Mft1-Thp2 pairs intersect at the vertex of the chevron forming a hydrophobic dimerization core (*Figure 4A*). An additional smaller homodimerization interface is mediated by the N-terminal helices of the two Tex1 molecules (*Figure 4B*). These two homodimerization interfaces appear to be the pivot point around which the rest of the Mft1-Thp2 coiled coils and the attached Tho2-Hpr1 platforms swing with a

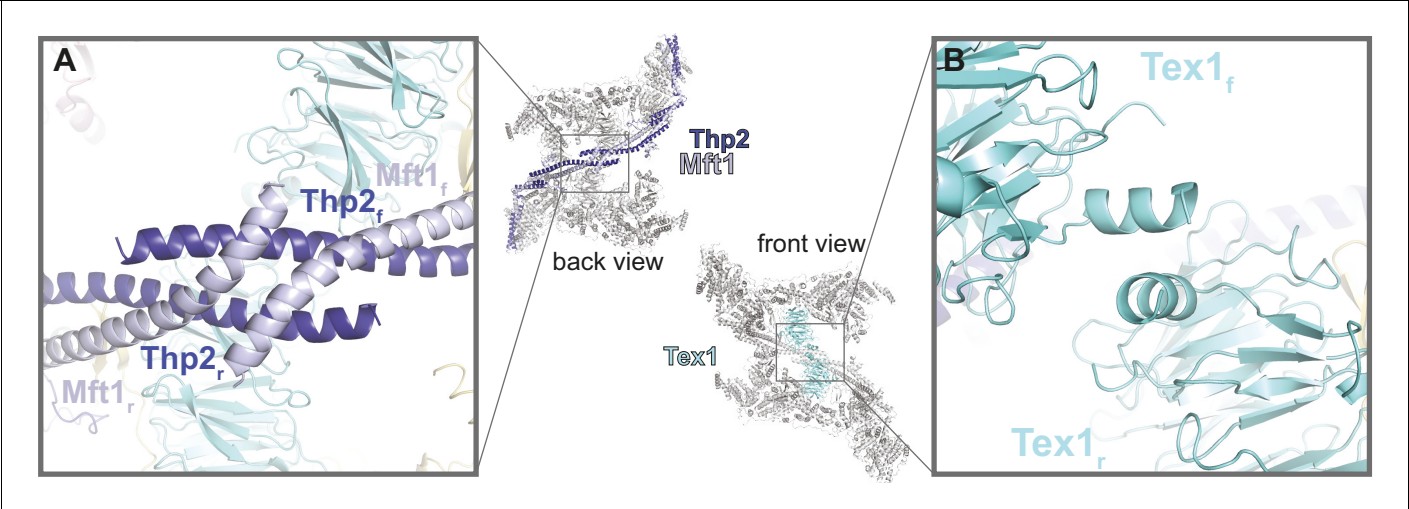

**Figure 4.** THO homodimerization properties. The central panel shows the back and front views of the THO-Sub2 homodimer, with the whole complex in gray except the dimerization elements highlighted in color: the two Mft1-Thp2 protomers (back view) and the two Tex1 protomers (front view). (A) Zoom-in view of the dimerization interface between the C-terminal coiled-coil portions of Mft1 and Thp2 rigid (r) and flexible (f) protomers (back view of the complex). (B) Zoom-in view of the dimerization interface between the N-terminal helices of the two Tex1 protomers (front view of the complex). The online version of this article includes the following figure supplement(s) for figure 4:

**Figure supplement 1.** Superposition of the two THO protomers.

seesaw-like movement (*Video 1*; *Figure 2D*). Variance analysis of the cryo-EM data shows that the movement is connected to changes in the two Sub2 proteins.

## THO dimer asymmetry is connected to different conformations of two Sub2 ATPases

Like other members of the DEAD-box family of ATPases, Sub2 is expected to undergo conformational changes connected to the nucleic-acid unwinding cycle (*Ozgur et al., 2015b*). In the ATP-RNA-bound active state, the two RecA domains adopt a well-defined closed conformation with a deep ATP-binding crevice and a shallow single-stranded RNA-binding cleft containing a sharp bend that is thought to be important for local RNA unwinding (*Ozgur et al., 2015a*; *Ren et al., 2017*). In the inactive state, DEAD-box proteins generally adopt more variable conformations in which the two RecA domains are spatially restrained by the flexible linker connecting them (*Ozgur et al., 2015b*). Members of the DEAD-box family (eIF4A, Dbp5, and DDX6) have also been shown to adopt an intermediate state upon binding MIF4G-like regulators, which maintain the RecA domains in a semi-closed (activated) conformation that resembles the structure adopted in the active state (*Mathys et al., 2014*; *Montpetit et al., 2011*; *Schütz et al., 2010*).

In the THO-Sub2 cryo-EM reconstruction, the Sub2 protomer at the proximal side of the homo-dimer adopts an intermediate semi-closed conformation, similar to that observed in other DEAD-box proteins in the activated state (*Mathys et al., 2014*; *Montpetit et al., 2011*; *Schütz et al., 2010*; *Figure 5* and *Figure 5—figure supplement 1*), rationalizing the Sub2 ATPase-activating properties of THO (*Ren et al., 2017*). The interactions that keep Sub2 in the activated conformation appear to be more extensive as compared to those in other DEAD-box proteins, as Sub2 binds at three sites in the 'body' of the rigid protomer (module-4, -5, and the latch; *Figure 5*) and appears to be additionally stabilized by contacts with the 'head' of the flexible protomer. These contacts are not present at the distal side of the homodimer, where the distance between the two protomers is larger and where indeed there is a higher degree of flexibility in the density map (*Figure 2D*). Thus, in our cryo-EM reconstruction of a THO complex in a resting state (e.g. without substrate or energy source), one Sub2 is kept in an activated conformation at the proximal side and the other Sub2 is in a flexible conformation at the distal site. Conformational changes of the Sub2 RecA domains upon substrate binding are expected to trigger changes in the connections with which the modules are

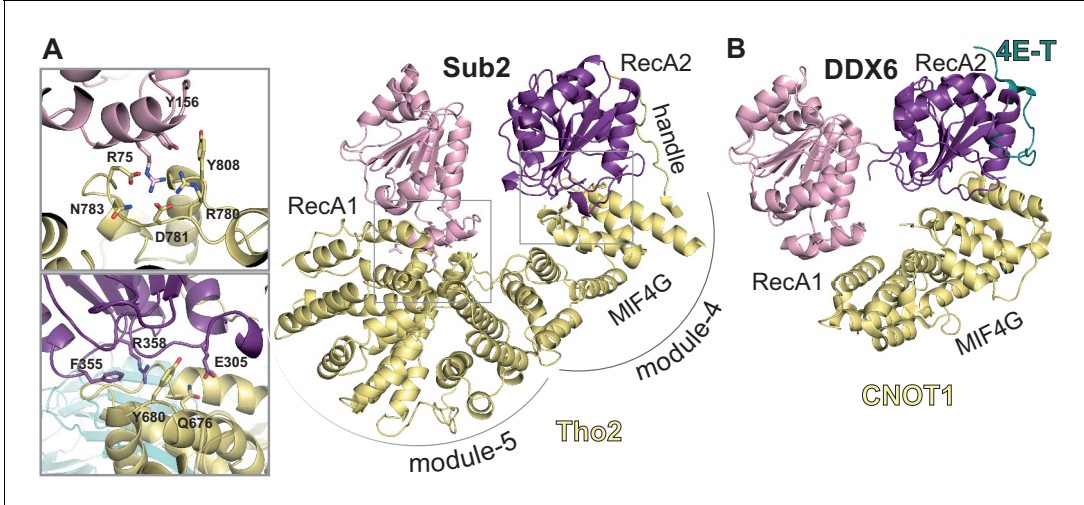

**Figure 5.** Sub2-activated conformation at the proximal side of the THO homodimer. (**A**) Sub2-Tho2 interaction at the proximal side. The zoom views show a subset of conserved interacting residues. See also *Figure 3—figure supplements 2* and *5*. (**B**) Structure of DDX6-CNOT1-4ET (*Ozgur et al., 2015a*) shown in the same orientation as Sub2-Tho2 in panel A after superposition of their RecA2 domains. Note that the Tho2 'handle' binds RecA2 at the equivalent position as protein 4E-T.

The online version of this article includes the following figure supplement(s) for figure 5:

**Figure supplement 1.** Conformational states of Sub2.

sequentially latched on to each other, thus imparting directionality to the intrinsic seesaw-like movements that we observed in the resting state of the complex.

## Mechanistic model for THO/TREX function

In all DEAD-box proteins, the active ATP consumption step depends on the concomitant presence of RNA (*Linder and Jankowsky, 2011*; *Zingler et al., 2008*). In addition, Sub2 requires the last subunit of the TREX complex, Yra1, to efficiently bind RNA (*Ren et al., 2017*). Yra1 has an unusual domain architecture with a central RRM (RNA recognition motif) domain connected by low-complexity sequences to similar and conserved motifs at the N- and C-termini (N-box and C-box motifs): the C-box specifically binds the Sub2 RecA1 domain while the N-box can also bind Sub2 with a similar affinity (*Figure 6—figure supplement 1A*; *Ren et al., 2017*; *Strässer and Hurt, 2001*; *Stutz et al., 2000*). Given the dimeric nature of THO-Sub2, a single Yra1 molecule could thus, in principle, bridge the Sub2 molecules at the proximal and distal sides of the complex (*Figure 6* and *Figure 6—figure supplement 1*). We envision that the RNA-dependent ATP binding and hydrolysis steps of Sub2 allow the complex to travel along the nascent mRNA and prevent the formation of aberrant secondary structures. It is also possible that the Sub2 ATPase cycle may impact the loading of Yra1 onto the mRNA.

The dimeric nature of THO-Sub2 also has important implications for how the complex may function in preventing R loop formation. Like all other helicases, Sub2 binds an RNA strand with fixed polarity, namely with the 3' end at RecA1 and the 5' end at RecA2 (*Ren et al., 2017*). In addition, the human Sub2 orthologue UAP56 has been shown to bind and unwind RNA:DNA duplexes (*Pérez-Calero et al., 2020*). Since the two Sub2 proteins are kept in opposite orientation by the two antiparallel Tho2-Hpr1 platforms in the THO dimer, the interacting nucleic acid strands would in turn need to have an opposite orientation. The topology of Sub2 within the complex is therefore compatible with the topology of R-loops, where the two strands of the RNA:DNA hybrid have opposite

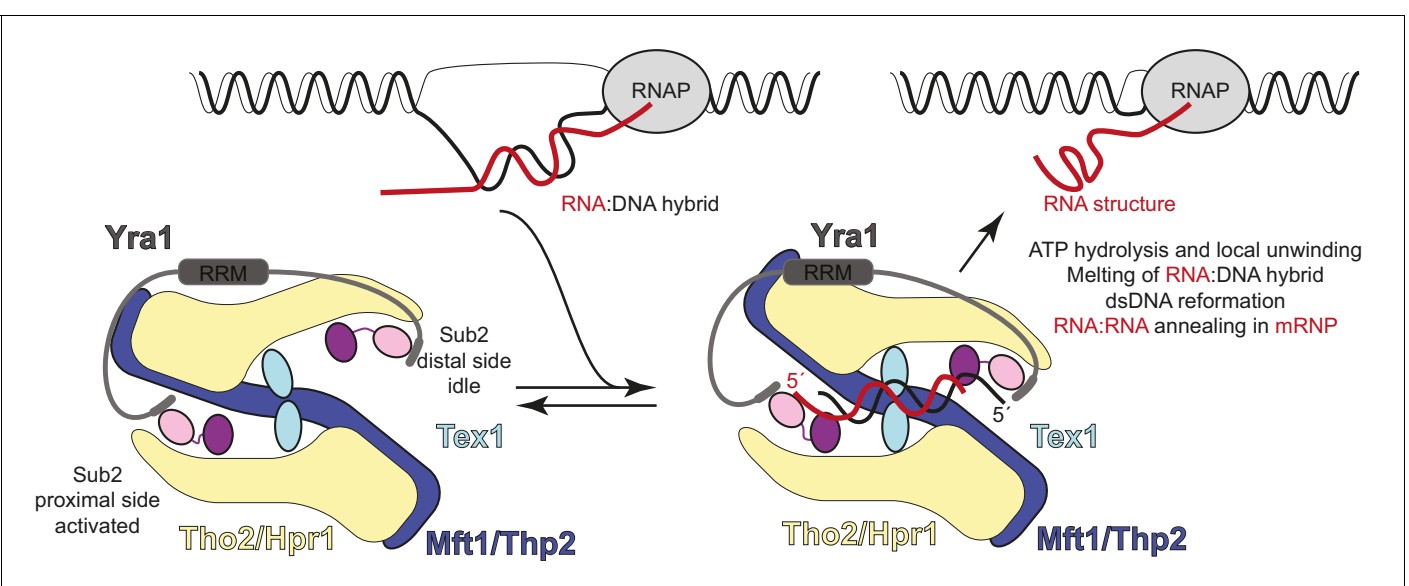

**Figure 6.** Hypothetical model of transcription-export (TREX) molecular mechanisms. Schematic depicts the TREX complex: on the left at resting state, with Yra1 bound to the Sub2 RecA1 domains via its N-box and C-box motifs (*Ren et al., 2017*); on the right in a substrate-binding state, with an RNA:DNA hybrid positioning the 5' ends of the RNA (red) and DNA (black) strands at the two opposite RecA1 domains (see also *Figure 6—figure supplement 1*). Binding of the RNA strand to the activated Sub2 would require changes in their relative orientation. In this hypothetical model, the energy released at the RNA-dependent ATP hydrolysis step is harnessed in a mechanical movement as the separated strands are dissociated and the complex returns to the resting state. The involvement of such mechanical force explains how incorporation in the complex may allow Sub2 to resolve RNA:DNA hybrids that would otherwise be too long to be melted by a DEAD-box protein in isolation (*García-Pichardo et al., 2017*; *Linder and Jankowsky, 2011*).

The online version of this article includes the following figure supplement(s) for figure 6:

**Figure supplement 1.** Hypothetical model of THO-Sub2-Yra1 (transcription-export) in binding RNA:DNA hybrids.

polarity (*Figure 6* and *Figure 6—figure supplement 1*). We propose that after the RNA:DNA hybrid is recognized and melted by binding to the two Sub2 molecules in the complex, the individual strands are released upon ATP hydrolysis and can engage in appropriate interactions due to the vicinity of the other DNA strand and of the RNA-annealing activity of Yra1 (*Figure 6*). Such a mechanism reconciles the intrinsic duality of the TREX complex in mRNA biogenesis, to dissolve RNA:DNA hybrids that can form upon mRNA synthesis, thus preventing R-loop formation, and to favor the annealing of RNA:RNA structures, thus chaperoning the formation of mature mRNPs.

# Materials and methods

## Key resources table

| Reagent type (species) or resource | Designation | Source or reference | Identifiers | Additional information |
|---|---|---|---|---|
| Strain, strain background (*Escherichia coli*) | BL21 Star (DE3) pRARE | EMBL Heidelberg Core Facility | | Electrocompetent cells |
| Cell line (*Spodoptera frugiperda*) | IPLB-Sf21-AE | Gibco | | |
| Cell line (*Trichoplusia ni*) | BTI-Tn-5B1-4 | Gibco | | |
| Strain, strain background (*Saccharomyces cerevisiae*) | BY4741 (MATa) yeast | Euroscarf | Y00000 | |
| Strain, strain background (*Saccharomyces cerevisiae*) | BY4743 (MATa/α) yeast | Euroscarf | Y20000 | |
| Antibody | Anti-protein-A IgG (mouse, monoclonal) | Sigma Aldrich | P2921 | (1:333 dilution) |
| Recombinant DNA reagent | Tho2 | This paper (Materials and methods) | Uniprot P53552 | pFastBac Hta-Tho2 Conti Lab |
| Recombinant DNA reagent | Hpr1 | This paper (Materials and methods) | Uniprot P17629 | pFastBac Hta-Hpr1 Conti Lab |
| Recombinant DNA reagent | Mft1 | This paper (Materials and methods) | Uniprot P33441 | pFastBac Hta-Mft1 Conti Lab |
| Recombinant DNA reagent | Thp2 | This paper (Materials and methods) | Uniprot O13539 | pFastBac Hta-Thp2 Conti Lab |
| Recombinant DNA reagent | Tex1 | This paper (Materials and methods) | Uniprot P53851 | pFastBac Hta-Tex1 Conti Lab |
| Recombinant DNA reagent | Sub2 | This paper (Materials and methods) | Uniprot Q07478 | 3C-GST-fusion Conti Lab |
| Recombinant DNA reagent | GFP-Sub2 | This paper (Materials and methods) | | Conti Lab |
| Commercial assay or kit | Bac-to-Bac Baculovirus Expression System | ThermoFisher Scientific | | |
| Software, algorithm | SerialEM | https://bio3d.colorado.edu/SerialEM/ | SerialEM_3-8-0beta8_64 & SerialEM_3-8-0beta11_64 | |
| Software, algorithm | Focus | https://focus.c-cina.unibas.ch/wiki/doku.php | v 1.1.0 | |
| Software, algorithm | cryosparc | doi: 10.1038/nmeth.4169 | Cryosparc2 | |
| Software, algorithm | CTFfind4 | doi: 10.1016/j.jsb.2015.08.008 | | |
| Software, algorithm | TOPAZ | doi: 10.1038/s41592-019-0575-8 | | |
| Software, algorithm | UCSF Chimera | UCSF, https://www.cgl.ucsf.edu/chimera/ | | |

*Continued on next page*

*Continued*

| Reagent type (species) or resource | Designation | Source or reference | Identifiers | Additional information |
|---|---|---|---|---|
| Software, algorithm | UCSF ChimeraX | UCSF, https://www.rbvi.ucsf.edu/chimerax/ | | |
| Software, algorithm | COOT | http://www2.mrc-lmb.cam.ac.uk/personal/pemsley/coot/ | 0.9 | |
| Software, algorithm | Phenix | https://www.phenix-online.org/ | PHENIX 1.18 | |
| Software, algorithm | Molprobity | Duke Biochemistry, http://molprobity.biochem.duke.edu/ | | |
| Software, algorithm | PyMol 2 | PyMOL Molecular Graphics System, Schrodinger LLC | PyMOL 2.1 | |

## Biochemical reconstitution

*S. cerevisiae* Tho2 (1597 aa, 184 kDa), Hpr1 (752 aa, 88 kDa), Mft1 (392 aa, 45 kDa), and Thp2 (261 aa, 30 kDa) were co-expressed as full-length proteins in insect cells using the pFastBac system. Tex1 (422 aa, 47 kDa) was C-terminally truncated (residues 1–380) to increase protein stability. The proteins were tagged with an N-terminal TEV-cleavable His tag. Cells were infected with 1% (v/v) virus and harvested 72 hr after infection. The cells were pelleted, resuspended, and lysed with a Dounce homogenizer in lysis buffer containing 50 mM Tris-HCl (pH 7.5), 500 mM NaCl, 5% glycerol, 5 mM β-mercaptoethanol, and 30 mM imidazole, supplemented with complete protease inhibitor (Roche) and benzonase. THO complex was affinity purified using nickel-based affinity chromatography (IMAC, HIS-Select resin from Sigma-Aldrich). After washing with 20 column volumes (CVs) of lysis buffer, the bound THO complex was eluted by increasing the imidazole concentration to 300 mM. For tag-cleavage TEV protease was added and the complex was dialyzed overnight in 50 mM Tris-HCl (pH 7.5), 250 mM NaCl, 5% glycerol, and 5 mM β-mercaptoethanol. For tag-removal the cleaved complex was once passed over IMAC beads and the beads were washed with an extra CV of dialysis buffer. As a final purification step the complex was concentrated with an Amicon 30 kDa cut-off filter and purified via size-exclusion chromatography (Superose6, equilibrated in 25 mM HEPES [pH 7.5], 150 mM NaCl, 5% glycerol, and 2 mM DTT).

Sub2 was expressed as an N-terminal 3C cleavable GST-fusion protein in *E. coli* STAR pRARE cells. After cell lysis by sonication and pelleting of cell debris the supernatant was incubated with GSH affinity beads in 50 mM Tris-HCl (pH 7.5), 500 mM NaCl, 5% glycerol, and 2 mM DTT, supplemented with benzonase and AEBSF. After removal of the GST tag via 3C protease with on-column cleavage, the proteins were further purified via a heparin column and injected onto a size-exclusion chromatography column (Superdex200 16/60) equilibrated in 20 mM HEPES pH 7.5, 200 mM NaCl, and 2 mM DTT. Purified proteins were concentrated, flash frozen in liquid nitrogen, and stored at −80°C. GFP-Sub2 was purified according to the same protocol.

The analytical size-exclusion chromatography to assess the reconstitution of THO and THO-Sub2 shown in *Figure 1A* was carried out on a Thermo Scientific Vanquish Ultra High-Performance Liquid Chromatography (UHPLC) platform, a system with improved separation and equipped with a fluorescence detector. Using this system, the reconstituted THO complex eluted at a molecular mass expected for a dimer. Since addition of Sub2 resulted only in a minor shift of the elution peak, the analytical experiment to confirm complex formation was carried out with GFP-Sub2 and monitored at the appropriate wavelength showing that Sub2-GFP indeed co-migrated with the THO complex. The same reconstitution conditions with untagged Sub2 were used for the sample subjected to cryo-EM single particle analysis.

## Native complex isolation

A sequence coding for a C-terminal Twin-Strep-3C-Protein-A tandem affinity tag was inserted into the endogenous HPR1 locus in *S. cerevisiae* strains BY4741 (MATa) and BY4743 (MATa/α) using standard yeast genetics techniques. Yeast were grown in 500 ml YPD to OD600 = 1, harvested by filtration and frozen in liquid nitrogen until processing. Frozen yeast nuggets were lysed with a cryo-mill (SPEX), resuspended in 1 ml ice-cold purification buffer (50 mM potassium phosphate pH 8 and 0.1% NP40) and incubated for 30 min with protein-G dynabeads (Life Technologies) coated with

anti-protein-A IgG (Sigma-Aldrich). Beads were separated from the lysate with a magnet, washed four times in 1 ml purification buffer, resuspended in 200 µl purification buffer containing 250 ng 3C protease, and rotated for 30 min at 4°C. 3C-eluates were incubated for 30 min at 4°C with Mag-StrepXT magnetic beads (IBA) before washing beads three times in 1 ml purification buffer and eluting in 20 µl SDS-loading dye. Whole SDS-eluates were separated on 12% SDS-PAGE and stained with Instant Blue (Expedeon).

## Cryo-EM sample preparation and data collection

The THO-Sub2 complex was assembled by incubating a fivefold molar excess of Sub2 together with purified THO complex and separating the excess helicase using size-exclusion chromatography. Fractions containing the full complex were pooled and the complex stabilized by mild cross-linking using 1 mM BS3 (bissulfosuccinimidyl suberate) for 15 min at RT. The sample was quenched with ammonium bicarbonate and concentrated to 1 mg/ml using a 30 kDa cut-off table-top concentrator. For cryo-EM sample preparation, 4.0 µl of the purified complex were applied to glow discharged Quantifoil 2/1 grids, blotted for 3.5 s with force '4' in a Vitrobot Mark IV (Thermo Fisher) at 100% humidity and 4°C, and plunge frozen in liquid ethane cooled by liquid nitrogen.

Electron micrographs were acquired with a FEI Titan Krios transmission electron microscope (ThermoFisher) using SerialEM software (*Schorb et al., 2019*). Movie frames were recorded at a nominal magnification of 64,000 × (calibrated physical pixel size: 1.38 A°/px) using a K3 direct electron detector (Gatan) and a GIF quantum energy filter (Gatan) at 20 eV slit width. The total electron dose of approximately 55 electrons per A°$^2$ was distributed over 40 frames. Cryo-EM micrographs were processed on-the-fly using the Focus software package (*Biyani et al., 2017*). For Tho-Sub2, 6689 micrographs were collected.

## Cryo-EM data processing

The THO-Sub2 dataset was processed entirely in CryoSparc (*Punjani et al., 2017*). Dose-fractionated movies were gain-normalized, aligned, and dose-weighted using Patch Motion correction. The contrast transfer function (CTF) was determined using CTFfind4 (*Rohou and Grigorieff, 2015*). A total of 1000 particles were picked manually and used to train a model that was subsequently used to pick the entire dataset using TOPAZ (*Bepler et al., 2019*). A total of 922,935 candidate particles were extracted and cleaned using iterative-rounds of reference-free 2D classification. The 298,569 particles after 2D classification were used for ab initio model reconstruction with the SGD algorithm to prevent model bias. The particles were further iteratively classified in 3D using heterogenous refinement. The 113,076 particles belonging to the best-aligning particles were subsequently subjected to nonuniform 3D refinement, yielding 3.93 A° global resolution and a B-factor of −81.8 A°$^2$. Next, we carried out focused local refinement after signal subtraction on both protomers of the THO-Sub2 dimer. The reconstructions for the rigid and flexible protomers were both significantly improved, indicating a flexible dimeric interface. The flexible and rigid protomers yielded 4.01 A° with a B-factor of −73.6 A°$^2$ and 3.69 A° with a B-factor of −71.1 A°$^2$, respectively.

To further investigate the dynamic motion of the protomers with respect to each other we performed a variance analysis on three modes. For a cleaner visualization and to limit the influence of high-frequency noise, the resolution was filtered to 5 Å. Using the 'simple' output, a linear movie of 20 volumes along each mode was calculated. The volume series along the second mode is shown to contain the swiveling motion anchored at the Tex1-Tex1 interface of the THO-Sub2 dimer (*Video 1*).

The major heterogeneity within the dimer lies in the flexibility between the two protomers. Thus, we reanalyzed the dataset with a mask around the better-resolved protomer, yielding an improved map of the rigid protomer at 3.5 Å. Per-particle local CTF refinement improved the resolution to 3.4 Å with a temperature factor of −108.6 Å$^2$ after nonuniform refinement. This map was used for de novo model building of the THO complex.

## Model building

The reconstructed density was interpreted using COOT (*Emsley et al., 2010*). Model building was iteratively interrupted by real-space refinements using Phenix (*Liebschner et al., 2019*). Statistics assessing the quality of the final model were generated using Molprobity (*Chen et al., 2010*). Images of the calculated density and the built model were prepared using UCSF Chimera

(*Pettersen et al., 2004*), UCSF ChimeraX (*Goddard et al., 2018*), and PyMOL. The model we built de novo for the 5-subunit THO complex has very good stereochemistry (see Ramachandran plot, *Figure 2—figure supplement 4A*).

## Acknowledgements

We thank Daniel Bollschweiler and Tillman Schäfer at the MPIB cryo-EM facility; Claire Basquin for the fluorescence anisotropy experiments in *Figure 6—figure supplement 1*; Peter Reichelt for discussions on the yeast experiment in *Figure 1B*; and Steffen Schussler for support in purification. We thank all the members of the group for discussion and input, in particular Ingmar Schäffer and Christian Benda, and Courtney Long for editing the manuscript. This study was supported by funding from the Max Planck Gesellschaft, the European Commission (ERC Advanced Investigator Grant EXORICO), and the German Research Foundation (DFG, GRK1721, SFB/TRR 237) to E.C. This work was performed within the framework of SFB 1035 (German Research Foundation DFG, Sonderforschungsbereich 1035, Projektnummer 201302640, project A07) and Transregionaler Sonderforschungsbereich 237 (Projektnummer 369799452, project A08).

## Additional information

### Funding

| Funder | Grant reference number | Author |
| --- | --- | --- |
| European Commission | EXORICO | Elena Conti |
| Deutsche Forschungsgemeinschaft | 201302640 | Elena Conti |
| Deutsche Forschungsgemeinschaft | 369799452 | Elena Conti |
| Max Planck Society | | Elena Conti |
| German Research Foundation | DFG | Elena Conti |
| German Research Foundation | GRK1721 | Elena Conti |
| German Research Foundation | SFB/TRR 237 | Elena Conti |

The funders had no role in study design, data collection and interpretation, or the decision to submit the work for publication.

### Author contributions

Sandra K Schuller, Conceptualization, Investigation, Visualization, Writing - original draft; Jan M Schuller, Conceptualization, Formal analysis, Validation, Visualization, Methodology, Writing - original draft; J Rajan Prabu, Conceptualization, Resources; Marc Baumgärtner, Resources, Investigation; Fabien Bonneau, Conceptualization, Investigation, Methodology; Jérôme Basquin, Conceptualization, Supervision, Validation, Investigation, Methodology, Project administration; Elena Conti, Conceptualization, Supervision, Funding acquisition, Writing - original draft, Project administration

### Author ORCIDs

Sandra K Schuller (ID) https://orcid.org/0000-0002-1800-8014
Jan M Schuller (ID) https://orcid.org/0000-0002-9121-1764
J Rajan Prabu (ID) https://orcid.org/0000-0002-7726-9310
Fabien Bonneau (ID) http://orcid.org/0000-0001-8787-7662
Elena Conti (ID) https://orcid.org/0000-0003-1254-5588

### Decision letter and Author response

Decision letter https://doi.org/10.7554/eLife.61467.sa1
Author response https://doi.org/10.7554/eLife.61467.sa2

## Additional files

### Supplementary files

- Supplementary file 1. Cryo-electron microscopy data collection, refinement, and validation statistics.
- Transparent reporting form

### Data availability

Cryo-EM maps are available in the Electron Microscopy Data Bank (11859 and 11871). Atomic models are available in the Protein Data Bank (7APX and 7AQO).

The following datasets were generated:

| Author(s) | Year | Dataset title | Dataset URL | Database and Identifier |
|---|---|---|---|---|
| Schuller SK, Schuller JM, Prabu RJ, Baumgartner M, Bonneau F, Basquin J, Conti E | 2020 | yeast THO-Sub2 complex dimer | https://www.rcsb.org/structure/7AQO | RCSB Protein Data Bank, 7AQO |
| Schuller SK, Schuller JM, Prabu RJ, Baumgartner M, Bonneau F, Basquin J, Conti E | 2020 | yeast THO-Sub2 complex | https://www.rcsb.org/structure/7APX | RCSB Protein Data Bank, 7APX |
| Schuller SK, Schuller JM, Prabu RJ, Baumgartner M, Bonneau F, Basquin J, Conti E | 2020 | THO-Sub2 highRes | https://www.ebi.ac.uk/pdbe/entry/emdb/EMD-11859 | Electron Microscopy Data Bank, 11859 |
| Schuller SK, Schuller JM, Prabu RJ, Baumgartner M, Bonneau F, Basquin J, Conti E | 2020 | THO-Sub2 dimer | https://www.ebi.ac.uk/pdbe/entry/emdb/EMD-11871 | Electron Microscopy Data Bank, 11871 |

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
