## [Decision Letter]

**Acceptance summary:**

The manuscript describes the cryo-EM structure of the *S. cerevisiae* THO complex in complex with Sub2. This high-resolution structure of the THO-Sub2 complex is important for the field as it will be the basis for future mechanistic studies to unravel the function of this complex in co-transcriptional R-loop resolution and mRNA export.

**Decision letter after peer review:**

Thank you for submitting your article "Structural insights into the nucleic acid remodeling mechanisms of the yeast THO-Sub2 complex" for consideration by *eLife*. Your article has been reviewed by two peer reviewers, and the evaluation has been overseen by a Reviewing Editor and James Manley as the Senior Editor. The reviewers have opted to remain anonymous.

The reviewers have discussed the reviews with one another and the Reviewing Editor has drafted this decision to help you prepare a revised submission.

Summary:

The manuscript entitled, "Structural insights into the nucleic acid remodeling mechanisms of the yeast THO-Sub2 complex", by Schuller et al. describes the cryo-EM structure of the *S. cerevisiae* THO complex in complex with Sub2. As proposed previously, Tho2 and Hpr1 is shown to form a scaffolding platform of the complex. Based on the asymmetric orientation of the protomers the authors suggest a model for THO's activation of Sub2 as well as a mechanism for cellular co-transcriptional R-loop resolution. Although no biochemical experiments are performed to underline the functional significance of the structure or to test the validity of the proposed mechanism, this high-resolution structure of the THO-Sub2 complex is important for the field and will be of interest to a wide audience.

Essential revisions:

1) Without looking at the determined structure, the Abstract is hard to understand. The authors should try to clarify the model that is proposed.

2) TREX plays an important role in mRNA export and yet the model is focussed on R-loops and their resolution, giving the superficial impression this is the sole role of THO-sub2. Are the authors suggesting that all RNA extruded from RNA polymerase II adopts an R-loop conformation initially? Genome wide mapping does not obviously suggest this and there are sequence and genomic locus preferences for R-loop formation. Therefore, can the authors expand on how they think this complex might work with RNA not in an R-loop? Is it possible that ATP hydrolysis serves mainly to load Yra1p onto the RNA when an R-loop is not present?

3) In the Introduction, the authors should cite more recent reviews on mRNP biogenesis in *S. cerevisiae*.

---

## [Author Response]

Essential revisions:1) Without looking at the determined structure, the Abstract is hard to understand. The authors should try to clarify the model that is proposed.

We agree that the Abstract was hard to understand, as the model cannot be easily explained in the context of a short Abstract. We have rephrased the Abstract to: “THO subunits Tho2 and Hpr1 intertwine to form a platform that is bound by Mft1, Thp2, and Tex1. […] The overall structural architecture and topology suggest the molecular mechanisms of nucleic acid remodeling during mRNA biogenesis”.

2) TREX plays an important role in mRNA export and yet the model is focussed on R-loops and their resolution, giving the superficial impression this is the sole role of THO-sub2. Are the authors suggesting that all RNA extruded from RNA polymerase II adopts an R-loop conformation initially? Genome wide mapping does not obviously suggest this and there are sequence and genomic locus preferences for R-loop formation. Therefore, can the authors expand on how they think this complex might work with RNA not in an R-loop? Is it possible that ATP hydrolysis serves mainly to load Yra1p onto the RNA when an R-loop is not present?

We agree with the reviewer’s assessment. We did not intend to imply that the TREX complex is only involved in R-loop resolution or that all nascent RNA forms an R-loop. The text has been modified to better emphasize the role of TREX in mRNA export as well, both in the Introduction and in the Discussion. In particular, we added:

“We envision that the RNA-dependent ATP binding and hydrolysis steps of Sub2 allow the complex to travel along the nascent mRNA and prevent the formation of aberrant secondary structures. It is also possible that the Sub2 ATPase cycle may impact the loading of Yra1 onto the mRNA.”

3) In the Introduction, the authors should cite more recent reviews on mRNP biogenesis in S. cerevisiae.

Thank you for your comment. We have added more recent reviews to provide the most up-to-date published information detailing mRNP biogenesis in *S. cerevisiae*.